# Electrochemical Determination of Uric Acid Using a Nanocomposite Electrode with Molybdenum Disulfide/Multiwalled Carbon Nanotubes (MoS_2_@MWCNT)

**DOI:** 10.3390/nano14110958

**Published:** 2024-05-30

**Authors:** Johisner Penagos-Llanos, Rodrigo Segura, Amaya Paz de la Vega, Bryan Pichun, Fabiana Liendo, Fernando Riesco, Edgar Nagles

**Affiliations:** 1Departamento de Química de los Materiales, Facultad de Química y Biología, Universidad de Santiago de Chile (USACH), Santiago 9170002, Chile; johisner.penagos@usach.cl (J.P.-L.); amaya.pazdelavega@usach.cl (A.P.d.l.V.); bryan.pichun@usach.cl (B.P.); fabiana.liendo@usach.cl (F.L.); 2Facultad de Química e Ingeniería Química, Universidad Nacional Mayor de San Marcos, Lima 15081, Peru; fernando.riesco@unmsm.edu.pe

**Keywords:** molybdenum disulfide, multiwalled carbon nanotubes, uric acid, modified electrodes

## Abstract

This paper presents an application for a molybdenum disulfide nanomaterial with multiwalled carbon nanotubes (MoS_2_@MWCNT/E) in a modified electrode substrate for the detection of uric acid (UA). The modified electrode generates a substantial three-fold increase in the anodic peak current for UA compared to the unmodified MWCNT electrode (MWCNT/E). The MoS_2_@MWCNT/E surface was characterized by cyclic voltammetry (CV), scanning electron microscopy (SEM), energy-dispersive spectroscopy (EDS) and electrochemical impedance spectroscopy (EIS). The achieved detection limit stood at 0.04 µmol/L, with a relative standard deviation (RSD) of 2.0% (n = 10). The method’s accuracy, assessed through relative error and percent recovery, was validated using a urine standard solution spiked with known quantities of UA.

## 1. Introduction

UA is a product of purine metabolism in humans, which is the species with the highest levels of uric acid in urine and blood among mammals. Normal UA levels in adults are between 3.5 and 7.2 mg/dL [1]. Several reviews have been published about the effect of high levels of uric acid on people’s health. One of the first studies, from 2015, reports on the causes and treatment of gout, hyperuricemia, and the elimination of urate crystals [2]. Another review from 2017 deals with the pathogenic potential of uric acid, describing uric acid as a substance linked to diseases such as chronic inflammatory arthritis and gout and indicating that, as the predominant anti-oxidant molecule in plasma, uric acid is necessary for inducing immune responses [3]. In addition, a review conducted in 2021 addresses uric acid and cardiovascular diseases such as hypertension, atrial fibrillation, chronic kidney disease, heart failure and coronary artery disease [4]. These previous studies reveal the reasons for the current interest in UA and the need to develop new methodologies for its detection in biological samples. To this end, in 2020, a review explored the latest progress in the detection of UA as well as the history of the development of UA detection methods, which include spectral techniques (ultraviolet absorption, fluorescence), electrochemical approaches (voltammetry, electrochemiluminescence, surface plasmon resonance), chromatography (liquid and gas phase), capillary electrophoresis and isotope dilution mass spectrometry. One of the disadvantages of these techniques is their high cost [5]. Electroanalytical techniques have also begun to be used significantly in the last decade for the detection of UA due to their low cost. One of the first reviews published in 2011 concluded that the commonly most used substances for the development of electrochemical sensors for UA are nanoparticles, carbon nanotubes, polymers, and conducting polymers, which have detection limits very similar to those achieved by other, more expensive techniques [6]. A study from 2022 compares enzymatic and non-enzymatic electrodes for the detection of UA [7], in which the non-enzymatic electrode with the fewest reports is the one that uses MoS_2_ in the modification. Some reports use Al with MoS_2_ [8], graphene and indium tin oxide [9], poly(3,4-ethylenedioxythiophene) [10] or graphene [11], achieving detection limits between 1.1 and 0.14 µmol/L, respectively. In this context, MoS_2_ is a two-dimensional (2D) nanomaterial whose structure consists of two-dimensional S-Mo-S nanosheets chemically linked by van der Waals interactions [12]. Single-layer molybdenum disulfide (SLMoS_2_) or multilayer molybdenum disulfide (FLMoS_2_) were found, of which SLMoS_2_ has a specific surface area that improves conductivity [12]. In turn, FLMoS_2_ has spaces between layers that allow heavy metal cations to be intercalated, thus improving the adsorption and capture capacity of these cations [13]. FLMoS_2_ can also be employed in sonication-assisted exfoliation treatment with some solvents, generating SLMoS_2_ which has a large number of active sites that increase its electron transfer capacity, is difficult to oxidize, and has an ultrathin plane structure where electrons are confined to a plane of atomic thickness and therefore become sensitive to the environment [14]. In addition, MoS_2_ has been used in the modification of electrodes to detect organic and biological substances such as bovine albumin [15], methionine [16], sulfamethoxazole [17] and catechol [18] and metal cations such as Pb [19,20], As [21] and Hg [22].

The described characteristics of this nanomaterial and its recent use in the modification of electrodes make it a viable alternative for the detection of uric acid. This new method is also simple, sensitive and easy to manufacture, and it shows selective activity for UA compared to dopamine and ascorbic acid. This new application has been well characterized, and the activity for AU has been well verified, where its sensitivity, stability and accuracy are mainly highlighted.

## 2. Materials and Methods

### 2.1. Reagents and Instruments

All solutions used in the electrochemical cell and solution preparations were obtained from Mili-Q water (heal force superseries PW ultra-pure water). H_3_PO4, NaH_2_PO_4_ and Na_2_HPO_4_ were purchased from Merck (Darmstadt, Germany). DP, AA, UA, K_3_/K_4_Fe(CN)_6_ and KCl were purchased from Sigma-Aldrich (Darmstadt, Germany). The FLMoS_2_ nanoflake solution and MWCNTs were purchased from Graphene Supermarket (Ronkonkoma, NY, USA). The synthetic urine standard used was SigMatrix Serum Diluent, an aqueous solution containing 2% recombinant human albumin. Cyclic voltammetry (CV) and square wave voltammetry (SWV) ESI measurements were performed with a Chi-instruments 620-C (Austin, TX, USA), while energy-dispersive spectroscopy (EDS) was performed using an FEI QUANTA FEG 250 (Tokyo, Japan) with a secondary electron detector. An Ag/AgCl electrode and a platinum wire from Chi-instruments (Austin, TX, USA) were used as reference and auxiliary electrodes, respectively. An Elma S 10 H ultrasound bath was employed for the preparation of the modified electrode.

### 2.2. Measurements

The electrochemical cell was completed with 9.0 mL of ultra-pure water, 0.5 mL of PBS and 250.0 µL of UA 5.0 mmol/L (0.25 mmol/L in solution) by CV. For all SWV studies, between 10.0 and 100.0 µL of UA at 0.50 mmol/L (between 0.1 and 1.0 µmol/L in solution) was used. The validity of the EIS data and subsequent equivalent circuit modeling were assessed using the AfterMath data organizer (version 1.6.10523) software. Data validity was verified through the Kramers–Kronig test, with a stringent criterion and a chi-square value below 0.01. For the simulations, a Randles cell model was initially employed, with additional circuit elements incorporated iteratively to enhance the goodness-of-fit by at least one order of magnitude in the chi-square value.

### 2.3. Sample Treatment

The human urine standard SigMatrix Serum Diluent—an aqueous solution used to evaluate the accuracy of UA detection—was spiked with known quantities of UA and did not require prior treatment before analysis. Only 0.5 and 1.0 mL of the standard were used. This amount was necessary to obtain a more complex matrix.

### 2.4. Preparation of MWCNTs and MoS_2_@MWCNT/E

The FLMoS_2_ nanosheets were exfoliated by assisted sonication for 30 s. Subsequently, 100 µL of N,N-dimethylformamide (DMF) was added to a 300 µL aliquot of this solution and exfoliated again for another 30 s. Furthermore, 100 mg of MWCNTs were weighed in an agate mortar and then homogeneously mixed with the exfoliated MoS_2_ solution and 20 µL of mineral oil to form the composite. This was compacted into a PVC cylinder with a 1.0 mm diameter and 7.0 cm length, with an internal steel wire as a conductor.

## 3. Results and Discussion

### 3.1. Surface Electrode Characterization

The surface morphology of SWCNT/E and modified MoS_2_@MWCNT/E was studied by SEM, EDX and EIS. Figure 1a,b show the SEM images for the surfaces of MWCNT/E and MoS_2_@MWCNT/E. The inset of Figure 1c shows the energy-dispersive X-ray (EDX) mapping of MoS_2_@MWCNT/E at the S Kα1 and Mo Lα1 energies.

MWCNTs display a fibrous surface that has a typical diameter in the range of 50–100 nm (Figure 1a). MoS_2_ nanosheets are homogeneously distributed over the entire surface of the MWCNTs. EDX studies showed that the MoS_2_ nanosheets are composed solely of Mo and S with very few impurities (inset of Figure 1c). Furthermore, Mo and S are observed with a homogeneous distribution on the surface of the MWCNTs. From this analysis, it can be confirmed that the MoS_2_ nanosheets deposited on the MWCNTs maintain their structure.

EIS was employed to examine alterations in the electrode–electrolyte interface. Figure 2 illustrates Nyquist diagrams for both the MWCNT/E and MoS_2_@MWCNT/E composites. Due to the porous nature of the electrodes, a constant-phase element (Q) was used instead of capacitors [Magar 2021]. The equivalent circuit model (ECM) for MWCNT/E consists of a Randles cell, which includes the following components: the electrolyte resistance (R1), the charge transfer resistance (R2), the double-layer capacitance (Q1), and a second constant-phase element (Q2), which accounts for surface reactivity and diffusion processes. And the ECM for MoS_2_@MWCNT/E includes the following components: the resistance of the solution (R1), the electrolytic resistance through the higher-resistance layer (R2), the capacity of the higher-resistance layer (Q1), the charge transfer resistance (R3), the capacitance of the electrical double layer (Q2) between the solution and the porous layer and a constant-phase element that accounts for diffusion-controlled processes (Q3) (Appendix A), with their respective values shown in Appendix A. Surprisingly, the addition of MoS_2_ demonstrated a marked increase in the charge transfer resistance by almost eleven times, presented as a much larger semicircle when compared to that of MWCNT/E. This elevation can be attributed to its semiconductor properties, resulting in diminished conductivity compared to the inherently highly conductive pure MWCNTs [23,24]. This is in line with a previous report that showed a decrease in the charge transfer resistance upon MWCNT addition to MoS_2_ [25]. Owing to the electrode’s composition, the observed semicircle appears slightly elongated, prompting a representation as two semicircles in the equivalent circuit, while the absence of a clear separation suggests an embedded model. This proposition is further substantiated by the micrographs in Figure 1, which reveal the coexistence of MoS_2_ and nanotubes on the surface, suggesting intricate interactions between the electrolyte and both electrode constituents. However, the enhanced sensing performance and the higher electroactive surface area (ECSA) cannot be solely attributed to improved electrical conductivity. As previously reported, they are likely due to the electrocatalytic activity of MoS_2_ [8].

### 3.2. Electrochemical Characterization

The electroactive properties of MoS_2_@MWCNT/E were studied by CV. This technique allowed us to evaluate the active area surface. Figure 3 shows the cyclic voltammograms for a solution of 5.0 mmol/L of Fe(CN)_6_^3+/4+^ with 0.1 mol/L of KCl and the voltammograms for the same solution when varying the scan rate between 0.02 and 0.14 V/s. In Figure 3a, the presence of MoS_2_ nanosheets in the MWCNT electrode was observed to considerably increase the current of the Fe(CN)_6_^3+/4+^ redox system by more than two times. Furthermore, the Ip_a_/Ip_c_ difference was almost 1.0. In turn, the ΔV changed from 0.50 V to 0.30 V (red curve in Figure 3a). These results show that the presence of MoS_2_ nanosheets improved the electrocatalytic activity of the MWCNTs.

Figure 3b,c show plots of the effect of the anodic and cathodic peak currents as a function of the square root of the scan rate (ν)^1/2^. The results show that by proportionally increasing (ν)^1/2^ between 0.02 and 0.140 V/s, the anodic and cathodic peak currents increase proportionally for both electrodes. Furthermore, the values of the anodic and cathodic peak currents’ slopes were higher for MoS_2_@MWCNT/E. This increase in the currents indicates that the active surface area is larger in the modified electrode. The slope values, Ip(A)/(ν)^1/2^, are summarized in Table 1. The active area of the electrode in cm^2^ was calculated for both electrodes using the Randles–Ševčík equation, which is summarized in Table 1. The results show that the active area of the electrode modified with MoS_2_ increased by almost 100%, and this increase may be one of the causes of the changes in the potential and current of the Fe(CN)_6_^3+/+4^ system observed in Figure 3a. In the case of other electrodes modified with MoS_2_@MWCNTs and other substances such as 5-sulfosalicylic acid, a ΔV close to 0.10 V is reported for the Fe(CN)_6_^3+/4+^ system. There are very few reports on the studied electrodes modified with MoS_2_ and MWCNTs in the presence of Fe(CN)_6_^3+/4+^ that have calculated the active area value. In relation to the value calculated in this paper, the active area value is much smaller compared to other larger-diameter electrodes, in which the reproduced active area is normally greater than 0.10 cm^2^ in electrodes modified with metal cation oxides and nanoparticles [26].

### 3.3. Electrochemical Activity of UA with MoS_2_@MWCNT/E

Figure 4 shows the square wave voltammograms for UA using MWCNT/E and MoS_2_@MWCNT/E, where anodic peak currents were observed at 0.37 V with MWCNT/E and 0.33 V with MoS_2_ nanoflakes. This result indicates that the presence of MoS_2_ in the MWCNT electrode means less energy is required for the oxidation of UA. Additionally, the current increases by almost three times compared to the unmodified electrode. This increase may be due to the increase in surface area calculated in the previous section but also to the electrostatic interactions and biocompatibility between UA and MoS_2_, which allow the amount of UA on the electrode surface to be greater [9]. In relation to the value of the oxidation potential of UA, in other electrodes modified with MoS_2_ and conductive polymers such as PEDOT [10], carbon nanomaterials such as graphene [11] or Al foil [8], the observed potential is very similar, below 0.4 V. Meanwhile, with Au nanoparticles, the potential observed displays higher values of 0.40 V [27]. Therefore, in this new report, the modified electrode presents good activity and requires less energy to oxidize. The reaction can be attributable to the oxidation of the nitro group of AU (Figure 1) [28].

### 3.4. pH Effect

UA presents a pKa value of 5.4, exhibiting the behavior of a weak acid [29]. Therefore, the pH of the UA solution affects its oxidation–reduction behavior. To assess the effect of pH on the anodic peak currents and the anodic potentials’ peak of UA using MoS_2_@MWCNT/E, voltammetry measurements were developed at different pH values that ranged between 2.0 and 6.0 using PBS. The results are shown in Figure 5, where it is clearly observed that at more acidic pH values, the anodic currents are stronger (Figure 5a). It is possible that at pH values between 5.0 and 6.0, the currents are lower because these are pH values very close to the pKa value of UA; consequently, UA is chemically oxidized and competes with electrochemical oxidation. In addition, the potential values shift to more positive values when the pH values are more acidic, indicating that UA has a Nernstian behavior and that the potential depends on the pH value. The plot of Ep(V) vs. pH (Figure 5b) presented a negative slope value of −0.050 pH. This slope value is close to the theoretical value of 0.059 pH, indicating that the number of protons (H^+^) involved in the reaction is equal to the number of electrons (e^−^). Previous reports have shown almost the same slope value using other electrodes and suggested an equal stoichiometry of 2H^+^:2e^−^ [30]. In general, using electrodes modified with MoS_2_, the pH value reported with the highest value of anodic current for UA is almost neutral [10,11,27]. A pH value of 2.0 was chosen as optimal because the current is much higher.

### 3.5. Parameter Optimization

The parameters that directly influence the increase in the anodic current of UA, like the pH studied in Section 3.2, were the anodic peak currents, which increased from 0.02 to 0.05 µA when the pH values changed from 5.0 to 2.0, respectively. Additionally, the accumulation time (t_ACC_) and accumulation potential (E_ACC_) were studied in the deposit stage. Figure 5 shows the peak currents as a function of E_ACC_ (Figure 6a) and t_ACC_ (Figure 6b). It is clearly observed that the anodic peak current increases proportionally with an increase in E_ACC_ between 0.40 and 0.60 V. At higher potential values, the anodic peak currents decrease proportionally. A value of 0.6 V is close to the oxidation potential observed for UA. Therefore, at higher potential values, the current decreases due to UA undergoing electro-oxidation processes on the electrode surface in the deposit stage. Meanwhile, the optimal t_ACC_ was 50.0 s. At higher values, the current decreases proportionally, possibly due to the saturation of the electrode surface. A potential of 0.6 V for 50 s was chosen as optimal in the accumulation stage. In the stripping stage, the parameters’ frequency and pulse amplitude were 15.0 Hz and 0.025 V, respectively. With the optimized parameters of pH, _EACC_ and t_ACC_, the anodic peak current increased from 0.02 µA to 0.08 µA, an increase of almost 75% of the anodic peak current for UA.

### 3.6. Scan Rate Effect on UA Using MoS_2_@MWCNT/E

Studying the transport mechanism of the analyte on the electrode surface is crucial to calculating the useful life of the electrode because the adsorption processes limit the use of the modified electrode in several measurements due to the analyte adsorbed on the surface of the electrode undergoing memory processes. Additionally, diffusion-controlled processes allow for the use of the electrode in more measurements.

The mass transport mechanism was studied by CV, verifying the anodic peak current for UA as a function of the scan rate between 0.02 and 0.14 V/s. Figure 7 shows the cyclic voltammograms and the plots of I_p_ (µA) vs. the square root of the scan rate and log I_p_ (µA) vs. the log scan rate. The results show that the anodic current increases proportionally as the square root of the scan rate increases between 0.02 and 0.14 V/s, and the linear relationship presents an R^2^ of 0.991. Furthermore, the slope of the plot of log I_p_ (µA) vs. the log scan rate (V/s) has a value of 0.453. These results indicate that the process is controlled by diffusion. Repetition studies showed that the same electrode can be used up to 10 times without considerable changes in its activity. In this case, the difference between the first and the tenth measurement was 0.0018 µA, indicating a decrease of 1.8 × 10^−4^ µA for each measurement. The relative standard deviation (RSD) was 0.0001, and the coefficient of variation (%CV) was 2.0%.

### 3.7. Calibration Curve, Detection Limits and Reproducibility

The calibration curve was developed by SWV within the concentration range of 0.10–1.1 µmol/L of UA using MoS_2_@MWCNT/E and under the following optimal parameters: E_Step_: 0.01 V; E_AMP_: 0.025 V; frequency: 15 Hz, E_ACC_: 0.60 V; t_ACC_: 50.0 s, and PBS pH: 2.0. The voltammograms and calibration curves are shown in Figure 8. The current ranges between each standard are small due to the small electrode diameter of 1.0 mm. The detection limit is 0.04 µmol/L, which was compared with previous reports using electrodes modified with MoS_2_ nanoflakes, as summarized in Table 2. The detection limit reported in this work is equally acceptable compared to more complex electrodes.

### 3.8. Interference Study with DP and AA

Substances that may cause some type of interference with the UA anodic current were considered and evaluated. Substances such as glucose, proteins, ascorbic acid (AA), dopamine (DP) and heavy metal cations were evaluated. Only DP and AA showed activity with MoS_2_@MWCNT/E. The voltamperograms of UA, DP and AA are shown in Figure 9 The results show that DP is oxidized at a less positive power value close to 0.30 V (green curve). In turn, the UA signal (red curve) presents an increase in the anodic current when AA is added (blue curve). This result indicates that AA is oxidized at the same potential value observed for UA, but the increase is smaller compared to the increase observed for UA alone. Furthermore, an increase in the background current (blue curve) was observed that makes the increase in the anodic current of UA appear greater, but in reality, the increase in the anodic current for UA was 10.0% in the presence of AA. These results indicate that the detection of UA is better without the presence of AA.

### 3.9. Analytical Application

The actual utility of MoS_2_@MWCNT/E was evaluated using the urine standard synthetic SigMatrix Serum Diluent, an aqueous solution containing 2% recombinant human albumin (expressed in rice) in a phosphate buffered saline solution (pH 7.4) with 0.1% ProClin™ 950, spiked with 10.0 µmol/L of UA (Figure 10). The analyses were developed using the standard addition method. The average UA quantified was 11.2 ± 0.01 µmol/L, with a relative error (%ER) of 12.0%. In this test, measurements were performed with concentrations higher than those in Figure 8. Additionally, the UA potential value was observed at less positive values than those observed in previous studies, possibly due to a matrix effect in the sample. These results demonstrate the good accuracy of the new method.

## 4. Conclusions

In this new study, the authors present a new convenient method for the detection of UA biological samples using an electrode based on MoS_2_ nanosheets on MWCNTs through a simple modification process. The modified surface showed excellent electrochemical activity for UA, allowing for a detection limit below 0.10 µmol/L. The characterization of the surface was comprehensive. Several techniques, such as scanning electron microscopy (SEM), dispersive X-ray spectroscopy (EDS), electrical impedance spectroscopy (EIS) and cyclic voltammetry (CV), were employed to measure the electrochemistry behavior of the modified electrodes towards UA oxidation. The notable advantages of the MoS_2_-modified electrode are its ease of fabrication, reproducibility and stability. One of its few disadvantages is its low selectivity in the presence of ascorbic acid.

## Data Availability

Data are contained within the article.

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
