# Peer review of "Electrochemical Determination of Uric Acid Using a Nanocomposite Electrode with Molybdenum Disulfide/Multiwalled Carbon Nanotubes (MoS2@MWCNT)"

_nanomaterials, 2024, doi:10.3390/nano14110958_

Round 1

Reviewer 1 Report

Comments and Suggestions for Authors

1)     The manuscript adds little new knowledge to the field as Mo2S@MWCNT composite is not new and its application as the base of electrochemical applications has also been reported:

-Li, X., Zhang, J., Wang, R., Huang, H., Xie, C., Li, Z., ... & Niu, C. (2015). In situ synthesis of carbon nanotube hybrids with alternate MoC and MoS2 to enhance the electrochemical activities of MoS2. Nano letters15(8), 5268-5272.

-Yadav, A. K., Verma, D., & Solanki, P. R. (2021). Electrophoretically deposited L-cysteine functionalized MoS2@ MWCNT nanocomposite platform: a smart approach toward highly sensitive and label-free detection of gentamicin. Materials Today Chemistry22, 100567.

-Sahu, T. S., Li, Q., Wu, J., Dravid, V. P., & Mitra, S. (2017). Exfoliated MoS 2 nanosheets confined in 3-D hierarchical carbon nanotube@ graphene architecture with superior sodium-ion storage. Journal of materials chemistry A5(1), 355-363.

The level of distinctive novelty and originality have to be demonstrated.

2)   The response mechanism must be explained and demonstrated to justify the response and specificity for UA.

Author Response

The authors thank the reviewers because they help improve the quality of the work presented

Reviewer 1

1) The manuscript adds little new knowledge to the field as Mo2S@MWCNT composite is not new and its application as the base of electrochemical applications has also been reported:

-Li, X., Zhang, J., Wang, R., Huang, H., Xie, C., Li, Z., ... & Niu, C. (2015). In situ synthesis of carbon nanotube hybrids with alternate MoC and MoS2 to enhance the electrochemical activities of MoS2. Nano letters15(8), 5268-5272.

-Yadav, A. K., Verma, D., & Solanki, P. R. (2021). Electrophoretically deposited L-cysteine functionalized MoS2@ MWCNT nanocomposite platform: a smart approach toward highly sensitive and label-free detection of gentamicin. Materials Today Chemistry22, 100567.

-Sahu, T. S., Li, Q., Wu, J., Dravid, V. P., & Mitra, S. (2017). Exfoliated MoS 2 nanosheets confined in 3-D hierarchical carbon nanotube@ graphene architecture with superior sodium-ion storage. Journal of materials chemistry A5(1), 355-363.

The level of distinctive novelty and originality have to be demonstrated.

Reply: The authors agree with the observation. That is why the word new was not indicated in the title, it is just a new application, the novelty is that we have a micro electrode only as a carbon nanotube support, in most works carbon nanotubes are deposited on glassy carbon or mixed with paste carbon

2)   The response mechanism must be explained and demonstrated to justify the response and specificity for UA.

Reply: The reaction was added to the text, as scheme 1. since it is a previously well-known mechanism that has been well reported.

We explain the specificity from the point of view of a process of affinity and increase in surface area. compared to other substances such as ascorbic acid and dopamine, which are molecules that are oxidized at values very close to uric acid. These did not cause great interference, other substances were not necessary because their concentration in urine is very low and uric acid is in greater quantity. We have no other way to explain the interaction since it would be surface studies that are not the objective of this work.

Reviewer 2 Report

Comments and Suggestions for Authors

please see the attached document.

Author Response

The authors thank the reviewers because they help improve the quality of the work presented

Reviewer 2

The paper is interesting and well written. I recommend its publication after minor revision. Q1. Figure 2- the Y axe should be [- Z” (kΩ)]; In the legend, please indicate the solution composition and which was the applied potential for each electrode? In addition, in the main text, please comment about the meaning of the circuit elements: R1; R2; R3; Q1; Q2; Q3. Did you determine the charge-transfer resistance? How large was the value for each electrode?

Reply: Thank you for pointing out the error. We have made the necessary correction to Figure 2, where the Y-axis is now correctly labeled as [-Z” (kΩ)]. The legend has been updated to include the solution composition and the applied potential, which was 10 mV for each electrode.

Regarding the circuit elements, we have provided a detailed commentary in the main text, and a graphical representation has been included as supplementary material (Figure S1). The values for each of the circuit elements are also listed in supplementary material, specifically in Table S1.

Additionally, the charge-transfer resistance was determined to be 243 Ω for pure MWCNT and 2652 Ω for MoS2@MWCNT. This significant increase is now mentioned in the text, where we highlight "a marked increase in charge-transfer resistance, by almost eleven times."

Q2. Page 4; row 151 Please replace “…more than 200.0 %” with “….more than two times”.

Reply; changes were made to the text

Q3. Page 5; row 162 “…the Rander-Selvck equation” is NOT correct. It is the Randles-Ševčík equation!

Reply: changes were made to the text

Q4. Page 6; row 194 Please replace “….increases by almost 300% compared to the unmodified electrode” with “….increases by almost three times compared to the unmodified electrode.”

Reply: changes were made to the text

 Q5. Page 6; row 215 Please replace “… indicating that AU has a Nertesian behavior ….” with “….indicating that UA has a Nernstian behavior ….”

Reply: Reply; changes were made to the text

 In addition, please better emphasize the novelty of your paper in the Introduction section.

Reply: Reply; changes were made to the introduction

Round 2

Reviewer 1 Report

Comments and Suggestions for Authors

The authors' response is still not sufficient to accept the manuscript:

1)      Q1: The level of distinctive novelty and originality have to be demonstrated.

Reply: The authors agree with the observation. That is why the word new was not indicated in the title, it is just a new application, the novelty is that we have a micro electrode only as a carbon nanotube support, in most works carbon nanotubes are deposited on glassy carbon or mixed with paste carbon

This information should be highlighted to demonstrate the novelty and originality of the manuscript.

In any case, I wonder whether this novelty is sufficient to accept the manuscript for publication in Nanomaterials.

 2) Q2: The response mechanism must be explained and demonstrated to justify the response and specificity for UA.

Reply: The reaction was added to the text, as scheme 1. since it is a previously well-known mechanism that has been well reported.

The authors state that "the reaction may be due to oxidation of the nitro group of AU", which is expected, however no data are included to demonstrate:

-The increase in sensitivity due to the use of the MoS2@MWCNT electrode compared to the MWCNT electrode.

If the signal is due to oxidation of UA, any other compound with a similar Eox should interfere, as is the case for AA. Data should be added to justify that any other compound does not interfere.

-The method has been applied to the determination of UA in synthetic serum samples and, as the authors point out, the experimental conditions and the Eox are different due to the matrix effect. What new interferences appear under these conditions?

Author Response

The authors thank the reviewers because they help improve the quality of the work presented

Reviewer 2

This information should be highlighted to demonstrate the novelty and originality of the manuscript.

In any case, I wonder whether this novelty is sufficient to accept the manuscript for publication in Nanomaterials.

Reply: The novelty is related to a new electroanalytical methodology using nanomaterials. We try to highlight in the text that the novelty is the activity of the new material towards the oxidation of UA. The authors consider that it is sufficient justification to consider that this new application is very versatile.

The authors state that "the reaction may be due to oxidation of the nitro group of AU", which is expected, however no data are included to demonstrate:

Reply: The study was done with cyclic voltammetry and the data that justifies this fact is the observed potential value which we compare with other reports in the text indicating that the reaction is the same. Doing other studies to verify would distance us from the objective of this report.

-The increase in sensitivity due to the use of the MoS2@MWCNT electrode compared to the MWCNT electrode.

Reply: This effect is shown in Figure 4 and is discussed in section 3.3.

If the signal is due to oxidation of UA, any other compound with a similar Eox should interfere, as is the case for AA. Data should be added to justify that any other compound does not interfere.

Reply: For this reason, the authors used a urine standard that contains all the components of real urine but without UA. Real urine was not used because one of the universities sponsoring the project prohibits the use of real biological material. The interference of dopamine and AA was studied because these two substances are present with UA and are oxidized at the same potential value. Other substances were not studied because it is not possible to find them in the urine. On the other hand, what they can affect are cations, but these did not affect, this was indicated in section 3.5.

-The method has been applied to the determination of UA in synthetic serum samples and, as the authors point out, the experimental conditions and the Eox are different due to the matrix effect. What new interferences appear under these conditions?

Reply: There are no new interferences, only that the matrix of the synthetic serum contains substances that can be absorbed and compete with the adsorption of UA but it does not affect the signal, since it can be detected and quantified, with the serum no new oxidation signals of other substances appeared. , this is seen in figure 10

Round 3

Reviewer 1 Report

Comments and Suggestions for Authors

Accpet